# Neuropelveology: An Emerging Discipline for the Management of Pelvic Neuropathies and Bladder Dysfunctions through to Spinal Cord Injury, Anti-Ageing and the Mars Mission

**DOI:** 10.3390/jcm9103285

**Published:** 2020-10-13

**Authors:** Marc Possover

**Affiliations:** 1Possover International Medical Center, 8008 Zuerich, Switzerland; m.possover@possover.com; Tel.: +41-(0)-443872830; Fax: +41-(0)-443872831; 2University of Cologne, 50923 Cologne, Germany

**Keywords:** neuropelveology, LION procedure, genital nerves stimulation, chronic pelvic pain

## Abstract

Neuropelveology is a new specialty in medicine that has yet to prove itself but the need for it is obvious. This specialty includes the diagnosis and treatment of pathologies and dysfunctions of the pelvic nerves. It encompasses knowledge that is for the most part already known but scattered throughout various other specialties; neuropelveology gathers all this knowledge together. Since the establishment of the International Society of Neuropelveology, this discipline is experiencing an ever-growing interest. In this manuscript, the author gives an overview of the different aspects of neuropelveology from the management of pelvic neuropathic pain to pelvic nerves stimulation for the control of pelvic organ dysfunctions and loss of functions in people with spinal cord injuries. The latter therapeutic option opens up new treatments but also widens preventive horizons not only in the field of curative medicine (osteoporosis and cardio-vascular diseases) but also in preventive medicine and anti-ageing, all the way to future applications in the “Mars mission” project.

## 1. Introduction

In the 1990s, laparoscopy was introduced in the surgical treatment of pelvic cancers and deep endometriosis. The challenge then was to perform at least as well as in radically open surgery. The introduction of video-endoscopy allowed for perfect vision and a considerable improvement in the ergonomics of the laparoscopic surgeon, which was necessary for more complicated and longer procedures. Laparoscopic pelvic surgery has thus become an extensive and radical surgery with the consequence of the appearance of postoperative pain too often unexplained and neglected as well as often irreversible functional morbidities. Patients who presented to neurourologists and neurologists did not find much help, only neuroleptic treatments but without any effort to research or treat the cause of the symptoms. The term “minimally invasive surgery” thus became more and more paradoxical. The only possibility to reduce this morbidity seemed to be the in-depth study of the surgical anatomy of the pelvic nerves and their sparing as successfully as possible during interventions. However, although topographical anatomy is extensively described in anatomy textbooks, the operative functional anatomy of the pelvic nerves was, on the contrary, almost completely non-existent.

Incidences of pelvic nerves pathologies are widely underestimated because of a lack of awareness that such lesions may exist, a lack of diagnosis and acceptance and a lack of declaration and reporting of such lesions. The most probable reasons for the omission of the pelvic nerves in medicine are the complexity of the pelvic nerve system, the difficulties of etiologic diagnosis and—probably the overriding reason—the limitations of access to the pelvic nerves for neurophysiological explorations and neurosurgical treatments. Neurosurgical procedure techniques are well established in nerve lesions of the upper limbs but pelvic retroperitoneal areas and surgeries to the pelvic nerves are still unusual for neurosurgeons. Few open-surgical approaches to the sacral plexus have been described by neurosurgeons for the treatment for traumatic pelvic plexopathies but these approaches are laborious and invasive, offer only limited access to the different pelvic areas and expose patients to the risk of severe vascular complications. Techniques of nerve neuromodulation to control pelvic pain syndromes and dysfunctions are for the same reasons limited to spinal cord and sacral nerves roots stimulation that considerably restrict their indications and effectiveness.

The use of the endoscope in combination with neurofunctional surgical procedures to the pelvic nerves proved to be a decisive advantage in this development [1,2,3,4] and in fact it was the beginning of a new medical specialty, neuropelveology [5,6,7]. This specialty combines the knowledge required for a proper neurological diagnosis, which is essential for an adapted treatment for intractable pelvic neuropathies. The concept of neuropelveology, the first medical practice that focused on the pathologies of the pelvic nervous system, was introduced more than twenty years ago by Possover. Since then, neuropelveology has established itself as a specialty in its own right, promulgated by the creation of the International Society of Neuropelveology in 2014.

Neuropelveology presents three consecutive aspects; the diagnostic stage followed by the therapeutic stage and the post-therapeutic follow-up of the patient. It covers four major areas:

The diagnosis and treatment of pelvic neuropathic pain with particular new techniques of laparoscopic pelvic nerves decompression and neurolysis.

The treatment of pelvic organ dysfunctions, in particular the stimulation of the genital nerves (genital nerves stimulation (GNS) therapy).The technique of laparoscopic implantation of neuroprothesis to the pelvic nerves (LION procedure) for the recovery of the loss of functions in people with spinal cord injuries.The stimulation of the pelvic autonomic nervous system for the prevention and/or treatment of general medical conditions such as osteoporosis, some cardio-vascular disease or control of sarcopenia (process of ageing).

The diagnostic stage uses its own instruments and an anamnesis covering many aspects from gynecology, urology, orthopedics, pelvic vessel pathology and psychology of the chronic patient and parapleology. The clinical examination combines the examination of the pelvic organs and their functions, the neurological examination of the musculoskeletal system with a neuropelveological examination and the palpation of the pelvic nerves by the vaginal or rectal route [8]. As somatic, neuropathic pain is more specific, a neuropelveological workup typically allows for specific diagnosis of the lesion site in the pelvic nerves.

Neuropelveolgy encompasses various medical treatments and surgery of the pelvic nerves. The latter includes neurosurgical techniques ranging from decompression, neurolysis, reconstruction and even nerve resection (e.g., sciatic nerve endometriosis) to pelvic neurofunctional surgery.

## 2. Neuropelveology for the Management of Chronic Neuropathic Pelvic Pain

Chronic pelvic pain (CPP) is a common condition involving multiple, organ-specific medical specialties, each with its own approach to diagnosis and treatment. Its management requires a knowledge of the interplay between pelvic organ functions and neurofunctional pelvic anatomy and also of the neurological and psychological aspects. However, no current specialty field takes this approach into account. Neuropelveology is an emerging discipline focusing on the pathologies of the pelvic nervous system on a cross-disciplinary basis [7].

The neuropelveological approach to pelvic neuropathies is primarily diagnostic with the application of neurological principles and an absolute knowledge of the pelvic neurofunctional anatomy. Patient history is the key with a focus not simply on the pain location but also on pain history, irradiation, aggravating factors, vegetative and somatic symptoms. The first step is to evaluate whether the pain is visceral or somatic (Table 1).

Visceral pain by the lesion of the hypogastric plexuses is recognized due to the diffuse nature of pelveo-abdominal pain, irradiations proximal to the lower back and multiple vegetative symptoms including malaise, oppression, syncope, irritability, nausea, vomiting and fatigue. The clinical examination focuses on specific clinical details for vegetative disorders such as pupil dilation, salivation inhibition and tachycardia. In somatic pain, it is essential to adopt a neurological way of thinking since the location of the pain and the location of the etiology is mostly different. Somatic pain is located superficially at the skin and is described as allodynia or an electrical shock with a very specific location, caudal irradiations to the genito-anal areas or to the lower extremities (dermatomes) and lack of vegetative symptoms. The neuropelveological workup scheme follows these six steps:(1)Determination of the nerve pathways involved in the relay of pain information to the brain.(2)Determination of the location of the neurological irritation/injury (troncular vs. radicular vs. spinal vs. cerebral location).(3)Determination of the type of nerve(s) lesion: irritation vs. injury (neurogenic neuropathy).(4)Neurological confirmation of the suspected diagnosis by clinical examination with in particular the transvaginal or transrectal palpation of the pelvic somatic nerves with the reproduction of the trigger pain and Tinel’s sign (eventually with selective anesthetic nerve(s) blockade).(5)Determination of a potential etiology based on patient history and diagnostic imaging.(6)Corresponding etiology-adapted therapy.

It is absolutely crucial to understand which nerves are involved in the pain and then to assess whether it is a nerve irritation secondary to compression or whether it is an axonal nerve lesion. In the first instance, the neuropelveological treatment is based on the laparoscopic exploration/decompression; in the second, on the neuromodulation of the affected nerves.

The intervention in the area of the pelvic somatic nerves, which is covered by large vessels and a dense network of lymph nodes, has hitherto been hindered by the lack of minimally invasive surgical methods. However, developments in video-endoscopy enable the exploration of the retroperitoneal pelvic space with access to the lumbosacral plexus and possibilities for nerve decompression and neurolysis. The most frequent aetiologies treated in neuropelveology are:Sacral radiculopathy by vascular or fibrotic entrapment [9,10,11].Compression of the sacral plexus by hypertrophy or atypical insertion of the piriform muscle.Deep infiltrating endometriosis of the sacral plexus and the sciatic nerve [9,12].Tumor of the sacral plexus (Figure 1) [13,14] and post-surgical pelvic neuropathies [15,16].

This endoscopic approach further allows in the case of an axonal lesion for the laparoscopic implantation of neuroprosthesis (LION procedure) where electrodes are selectively placed in contact with the injured pelvic nerves for the possible control of neuropathic pain [17].

Post-therapy patient follow-up for pain management is essential. In nerve neuromodulation, the stimulation parameters must be calibrated at regular intervals. After laparoscopic nerve decompression, neuropathic pain first significantly increases while improvement usually does not set in until eight months after the operation. The follow-up of these patients is essential in order to adjust the medical treatment and to treat the pain-memory as successfully as possible. The latter, however, is much more difficult to direct.

## 3. Genital Nerves Stimulation (GNS) Therapy

Various sites have been used for the implantation of electrodes to the pelvic nerves to treat pelvic organ dysfunctions. Sacral nerve stimulation was the first technique for pelvic nerves stimulation that typically involves the electrical stimulation of the nerve via a dorsal transformational technique of implantation. Sacral nerve stimulation (SNS) and pudendal nerve stimulation evolved as a widely used treatment for an overactive bladder (OAB) but does not completely resolve symptoms in the majority of patients. Both techniques are still unusual for most gynecologists so that the field of pelvic nerve stimulation is still extremely restricted in gynecology. There is definitively a need for a more suitable alternative for neuromodulative treatments; methods that cannot only be reserved for experts in this field but for all gynecologists dealing in daily practice with patients suffering from functional disorders of the bladder. This is why the LION procedure of the sacral plexus [18,19] and then the pudendal LION procedure were developed [20]. However, both techniques of implantation remain too complex for the generalist gynecologist trained in surgery but not in neuropelveological procedures. The stimulation of the dorsal nerve of the penis/clitoris (DNP) emerges then as a very attractive alternative that might result in great outcomes for treating urinary and fecal disorders [21]. DNP is extremely interesting because its stimulation effectively increases bladder capacity, inhibits involuntary detrusor contractions and overactive bladder symptoms [22,23] and may even control idiopathic fecal incontinence [24].

Genital nerves stimulation (GNS) is the surgical procedure developed for the stimulation of the DNP, an implantation technique adapted to the most classical surgical approach in gynecology, the vaginal approach. The procedure consists of two phases: a preoperative non-surgical test-phase and a second phase involving the surgical implantation of the neuroprothesis. In contrast to the classical technique of stimulation, the GNS-test-phase is the only one which does not require any interventional procedure. Due to the fact that the genital nerves are located just a few millimeters below the skin’s surface, test-stimulation can be obtained using skin surface electrodes (Figure 2).

The effect of the stimulation can be tested by the patient in their daily, family and professional environment or alternatively at the practice under urodynamic testing or, if required, other electrophysiological testing.

After confirmation of the effectiveness of GNS, implantation of the permanent neuroprothesis can be scheduled. The procedure is performed either under general or spinal anesthesia or using only local anesthesia with IV sedation as in the classical tension-free vaginal tape procedure (TVT). The first step of the procedure consists of the introduction of a hollow curve needle applicator (Curve Applicator^®^ NeuroGyn AG, Baar, Switzerland) with a spear from below, behind the pubic bone according to the classical tension-free vaginal tape (TVT) procedure: A sagittal incision of about 2 mm in length is made approximately 1 cm below the external urethral meatus. The curve needle driver is inserted into the incision. The tip is oriented at an angle of 5–10° from the midline towards the symphysis. The inserter tip is approximately in the 11 o’clock position (1 o’clock on the right side). The curve needle driver is advanced, contacting the inferior edge of the pubic ramus, until it transfixes the urogenital diaphragm, enters into the retropubic space and comes out through the skin in the suprapubic area (Figure 3a–c).

The passage of the applicator behind and in direct contact with the dorsal aspect of the pubic bone is controlled with two fingers inserted into the vagina. A cystoscopy is performed to make sure the bladder and urethra are intact. The spear of the curve driver needle is removed. A quadripolar lead electrode with an electrode distance of 60 mm is introduced retrograde into the shaft of the curve needle driver; by the retraction of the curve needle driver, the electrode lead is left in position with the stimulation’s poles coming out through the vulvar incision (Figure 4a,b).

Through a second median supravulvar incision, the applicator with the spear is introduced from top to bottom so that it is as deep as possible (ventral to the pubic bone but as close as possible to it in order to assure the deep location of the cable electrode) and emerges through the first vulvar incision. After removing the spear, the electrode cable is inserted retrograde into the applicator again (Figure 5a–c).

To use the hollow needle driver for the retrograde introduction of the lead electrode enables the optimal placement of the lead electrode to the genital nerves without the need for any dissection, which, in turn, reduces considerably the risk of bleeding and nerve injury. The introduction of the curve needle driver from below belongs to standard urogynecology (TVT) procedure. As the (dorsal Nerve of The Penis/Clitoris) DNP perforates the perineal membrane laterally to the external urethral meatus at an average distance of 2.7 cm (2.4–3.0 cm) and then runs along the bulbous spongy muscle for a distance of 1.9 cm (1.8–2.2 cm) before penetrating the pillars of the clitoris (Figure 7), the second passage of the lead electrode in front of the pubis ensures direct contact of the electrode to the DNP [25].

The last step is then the connection of the lead electrode to the generator, which is finally fixed behind the pubic bone through a suprapubic mini-laparotomy. The fixation of the generator behind the pubic bone protects from external traumas and dislocation.

No X-ray screening, neurophysiological monitoring or stimulation with (Electromyography) EMG electrodes are mandatory during the procedure for a proper implantation. Due to the fact that the presented procedure does not need two surgical procedures for both the test and the final implantation but only one for the final implantation, the presented protocol allows a considerable cost reduction in comparison with the usual procedures for sacral or pudendal nerves stimulation.

## 4. LION Procedures

The endoscopic approach allows in case of axonal lesion or dysfunction of the nerves the selective laparoscopic implantation of neuroprothesis (LION procedure) for electrical stimulation of the nerves (Figure 8).

This procedure has been used for the treatment of nerve damage and pelvic organ dysfunctions as reported previously but probably the most impressive indication of this technique is the implantation in people with spinal cord injuries for the recovery of some walking functions [26]. In 2006, we performed the first LION procedure in a paraplegic patient for the control of the bladder function [27]. This intervention consisted of a laparoscopic implantation of a fine wire in direct contact with the endopelvic portion of the nerves for electrical stimulation [28]. Laparoscopic exposure of the endopelvic portion of both the sciatic and of the pudendal nerves was obtained by passing laterally to the external iliac vessels through the lumbosacral space and the gentle detachment of the inter-iliac lymph-fat-tissue from the pelvic sidewall to avoid lymphocele. Multiple channel electrodes enabled the stimulation of both nerves with only one lead electrode. Exposure of the femoral nerves was also obtained by the transperitoneal approach behind the major psoas muscle. The four lead electrodes were not fixed to the nerves (Figure 9) while the cables formed loops in the retroperitoneal space to avoid dislocation and were finally passed through the pelvic wall and connected to a rechargeable pulse generator implanted subcutaneously into the anterior abdominal wall.

Video: LION procedure in SCI (Appendix A)

The crucial discovery we made with the LION procedure in people with SCI was undoubtedly the fact that some patients experienced enough recovery of supra-spinal control for some leg movement or even standing and walking [26,29]. In the most recent study of 29 patients with SCI 10 years after a LION procedure, 20 of them (71.4%) were able to demonstrate an electrically assisted, voluntary extension of the knee [30] (Figure 10).

26 patients could get to their feet when the pacemaker was switched on (92.8%). Five patients could walk <10 m (17.85%) at the bar (Figure 11). Nineteen patients (AIS A: *n* = 8; AIS B: *n* = 9; AIS C: *n* = 2) could walk >10 m (67.8%); eight of them only at the bar (28.5%) and eleven of them with the aid of crutches/walker and without braces (40%).

The precise mechanism at work in people with SCI to recover walking functions after the LION procedure is still unknown. There is increasing evidence to suggest that neuromagnetic/electrical modulation promotes neuroregeneration and neural repair by affecting signaling in the nervous system but our findings suggest that the information signals to the brain might use not only anatomical nerve pathways but also functional pathways activated by a continuous low frequency stimulation of the low-motor neurons below the spinal cord lesion.

Beyond the psychological impact and the gaining of some autonomy, the benefits of locomotion include improvement of contractures, prevention of deep venous thrombosis and oedema and amelioration of spasticity [31]. Standing up in combination with gluteal muscle training (gluteal pads effect) protects patients from decubitus lesions, especially in the buttock [32]. Continuous low frequency stimulation of the implanted nerves outside periods of training may be advantageous for the reduction of spasticity [33] and the regulation of bone density [34,35]. Nerve stimulation has been reported in the treatment of arthritis of the legs [36] but also in vivo studies involving animal models have revealed that electric stimulation of wound healing processes results in more collagen deposition [37], enhanced angiogenesis [38], greater wound tensile strength [39] and a faster wound contraction rate [40]. In addition to these direct cellular actions, electrical stimulation has been shown to improve tissue perfusion and reduce oedema formation that results in a significant increase in transcutaneous oxygen pressures [41]. Therefore, the LION procedure to the pelvic nerves is potentially useful in the rehabilitation of people with spinal cord injuries by reducing the risks of complications.

The LION procedure to the pelvic somatic nerves has been further reported for treating urinary dysfunctions and improving locomotion in multiple sclerosis patients [42].

## 5. Future Visions in Neuropelveology: The “in-Body-ENS”

The development of new technologies to assist paraplegics with their common problems associated with inertia when confined to a wheelchair may find revolutionary applications in preventive medicine and even in the world of space missions in the future. The LION procedure enables a continuous and passive electrical nerve stimulation (ENS) without the need for an external stimulation system, while the neuroprothesis is located within the body: the in-Body-ENS. This capability of continuous in-body electrical nerve stimulation may open the door to a whole new area of humanity in which implanted electronics may help the human body to a better performance and a longer life. The process of ageing, also called sarcopenia, is characterized by muscle atrophy along with a reduction in muscle tissue quality characterized by such factors as the replacement of muscle fibers with fat and the degeneration of the neuromuscular junction leading to a progressive loss of muscle function and frailty. Prevention of the aging process mainly focuses on the control and treatment of such a muscle atrophy. Several therapies have been proposed for preventing the aging process such as mental activity, muscle training and high-protein diet. A crucial factor in this is sustaining a high individual strength capacity: The elderly need strength training more and more as they grow older to stay mobile for their everyday activities. The crucial factor in maintaining strength capacity is an increase in muscle mass. As continuous passive stimulation of the pelvic somatic nerves enables muscle training and may reduce the process of muscle atrophy, the in-Body-ENS may become an option in the future for slowing down the aging process by preserving body muscle mass. This technique may be appropriate in elderly people who are not capable of active muscle training because of pain, motoric limitations or subcortical pathologies but also in people confined to bed for long periods of time (prophylaxis of decubitus).

As sympathetic trunks travel downward outside the spinal cord and first anastomose to the sacral plexus, which build the sciatic nerve, continuous low frequency/low energy sciatic nerve stimulation (passive in-body- (Functional Electrical Stimulation) FES) permits neuromodulation of the sympathetic nervous system of the lower extremities and of the bottom. Due to the fact that there is further evidence of the role of the sympathetic innervation of bone tissue and of its role in the regulation of bone remodeling in humans, sympathetic nerve stimulation obtained by stimulation of the pelvic somatic nerves might also open new techniques for the prevention of osteoporosis not only in people with SCI as demonstrated in our study but also in elderly people [34,35].

In addition to this, the in-body-ENS may also find revolutionary applications in the world of space missions. Space is a dangerous, unfriendly place that requires daily exercise to keep muscles and bones from deteriorating. Calf muscle biopsies before flight and after a six month mission on the International Space Station show that even when crew members did aerobic exercise for five hours a week and resistance exercise three to six days per week, muscle volume and peak power both still deteriorated significantly. The in-body-ENS, by contrast, may allow muscle mass to be maintained even whilst the astronaut is at rest and provides an extremely effective and timesaving strength training program. During space flight, crew members also lose bone density; the calcium that is released ends up in the urine, which contributes to an increased calcium-stone forming potential. If the stone completely blocks the tube draining the kidney, the kidney could cease to function with catastrophic even life-threatening consequences for the astronaut. Due to the excruciating pain, affected astronauts could become incapacitated and missions may have to be aborted. Due to the fact that stimulation of pelvic sympathetic nerves may reduce this process of osteoporosis, as shown in our paraplegic study, in-Body-ENS may present a potential prophylactic for kidney stone formation in microgravity.

## Figures and Tables

**Figure 1 jcm-09-03285-f001:**
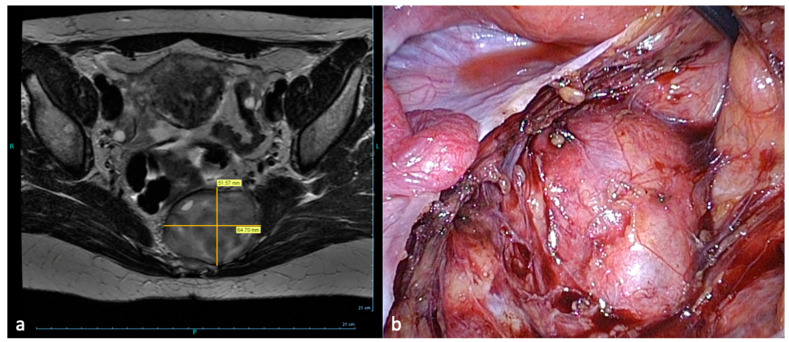
Laparoscopic resection of a sacral plexus schwannoma left. (**a**) MRI-presacral schwannoma (**b**) Corresponding intraoperative findings.

**Figure 2 jcm-09-03285-f002:**
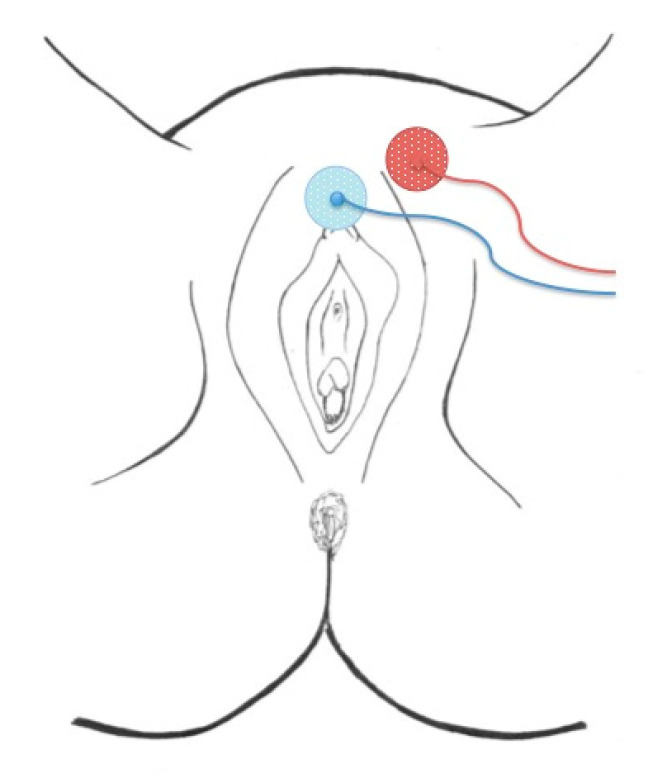
Position of the skin surface electrodes for the test-phase.

**Figure 3 jcm-09-03285-f003:**
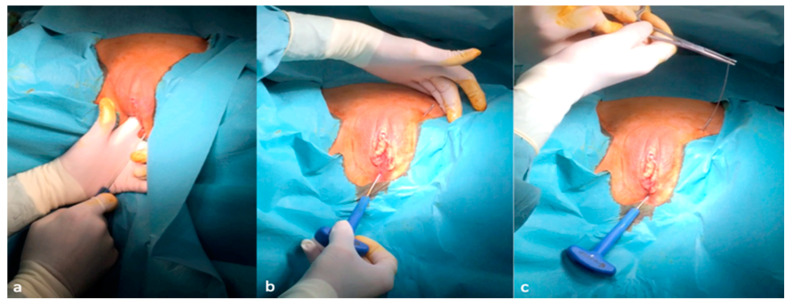
Introduction of the curve applicator from below and behind the pubic bone (**a**,**b**) and the removal of the spear of the curve driver needle (**c**).

**Figure 4 jcm-09-03285-f004:**
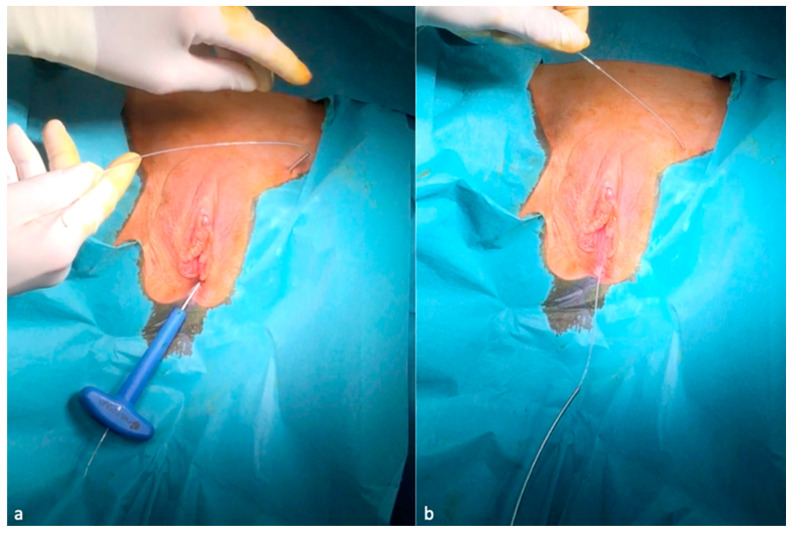
Introduction retrograde of the lead electrode (**a**) and the removal of the curve driver needle from below (**b**).

**Figure 5 jcm-09-03285-f005:**
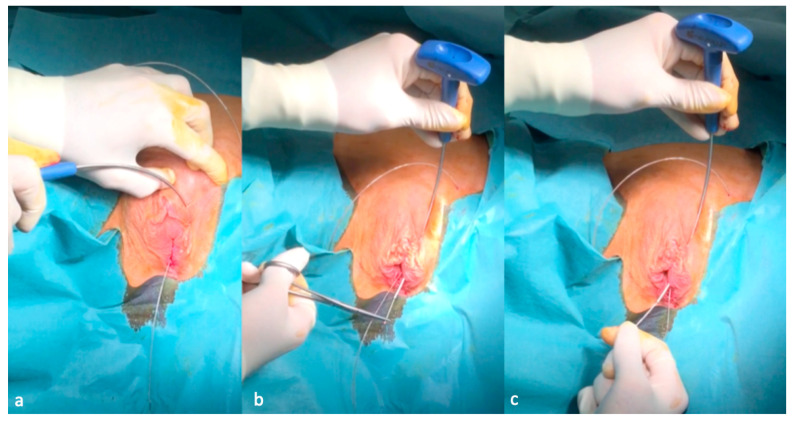
Second introduction of the curve applicator from above down to the first vaginal incision (**a**), removal again of the spear (**b**) and retrograde introduction once again of the lead electrode (**c**). After removing the applicator, the electrode is in place (Figure 6).

**Figure 6 jcm-09-03285-f006:**
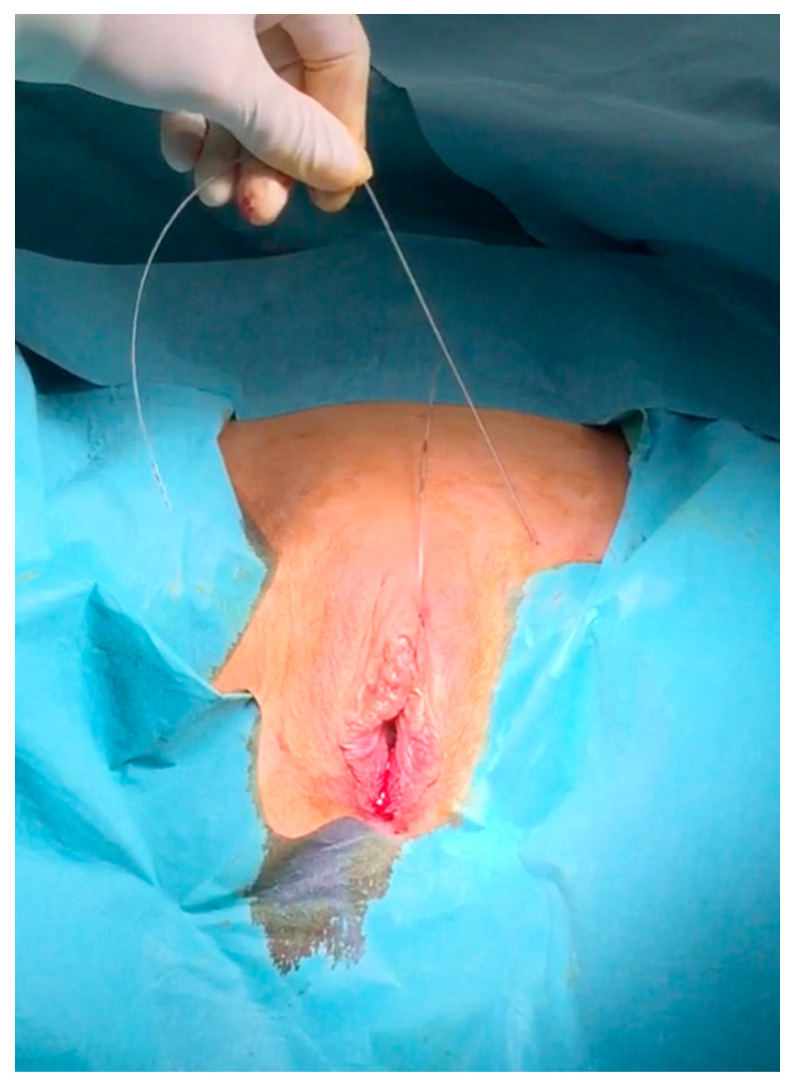
Retropubic passage of the lead electrode.

**Figure 7 jcm-09-03285-f007:**
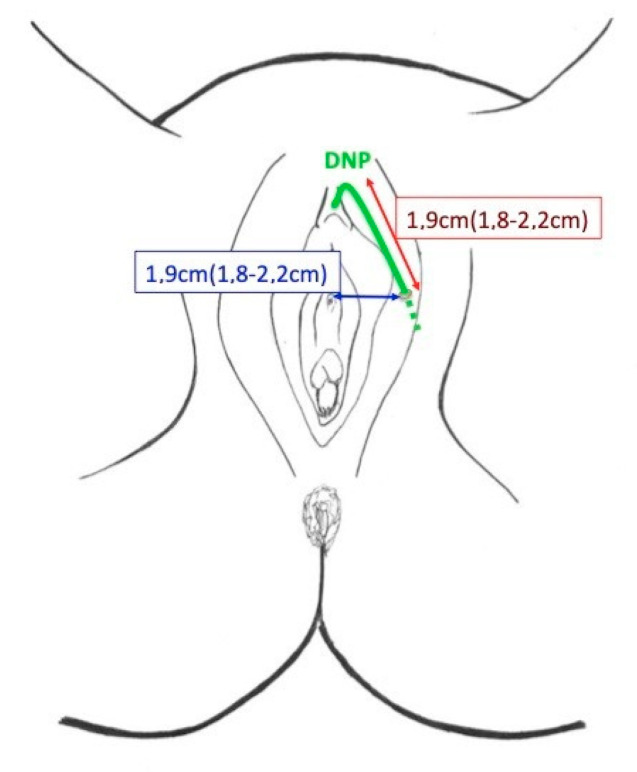
Dorsal nerve of the penis/clitoris (DNP) pathway at the vulva.

**Figure 8 jcm-09-03285-f008:**
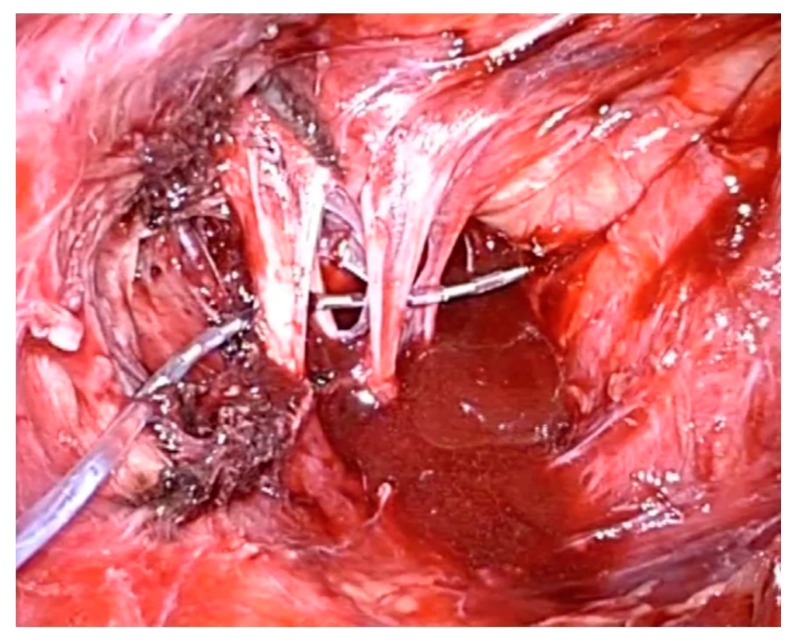
Sacral nerves laparoscopic implantation of neuroprothesis (LION) procedure.

**Figure 9 jcm-09-03285-f009:**
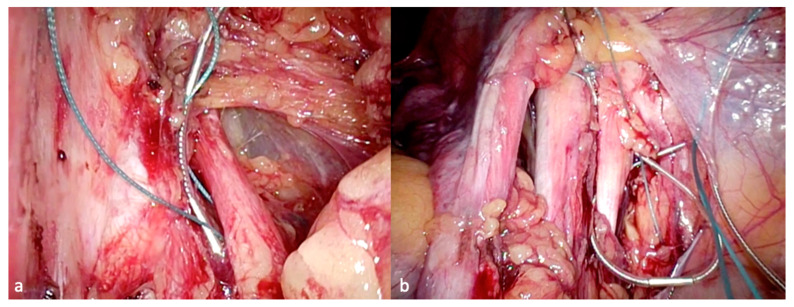
Placement of the lead electrodes to the left sciatic nerve (**a**) and the right femoral nerve (**b**).

**Figure 10 jcm-09-03285-f010:**
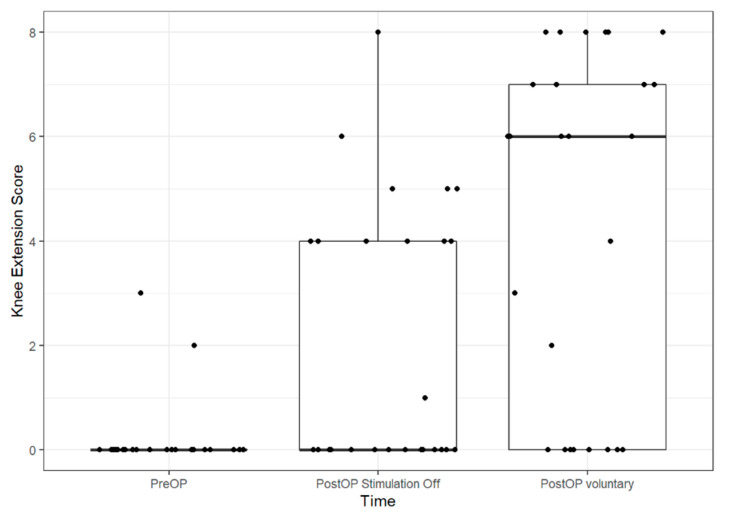
Boxplots of Knee Extension Score pre- and post-op. The line in the box shows the median, the lower and upper hinges correspond to the first and third quartiles and the upper/lower whisker extends from the hinge to the largest/smallest value no further than 1.5 IQR from the hinge (where IQR is the inter-quartile range). Dots show individual data points. PreOP-preoperative, PostOP-postoperative.

**Figure 11 jcm-09-03285-f011:**
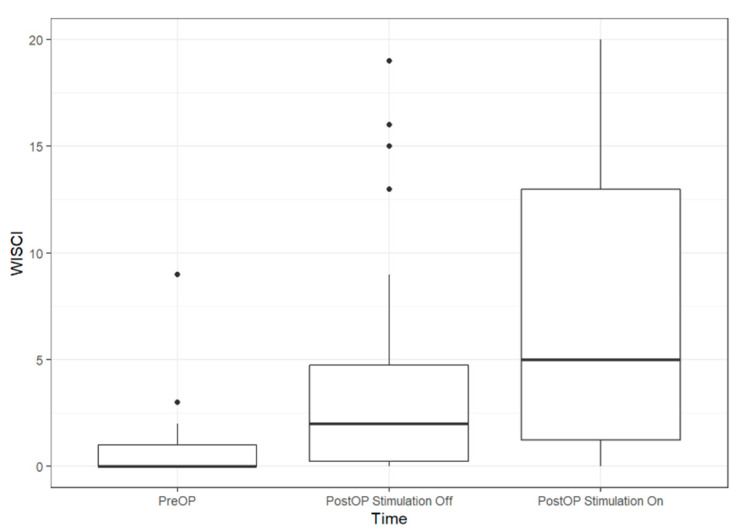
Boxplots of WISCI (Walking index for spinal cord injury) pre-operatively and at the 11/2018 follow-up. The line in the box shows the median, the lower and upper hinges correspond to the first and third quartiles and the upper/lower whisker extends from the hinge to the largest/smallest value no further than 1.5 IQR from the hinge (where IQR is the inter-quartile range).

**Table 1 jcm-09-03285-t001:** Visceral Versus Somatic Pain: Symptoms (out of: Possover M. Neuropelveology—latest developments in Pelvic Neurofunctional Surgery—ISBN: 97895244533-0-8, 2015:26).

Visceral Pain	Somatic Pain
Pain Quality: Vague; Poorly Localized in The Entire Lower Abdomen with Radiation Into The Lower Back; Dull in Nature.	Pain Quality: Allodynia; Similar to An Electrical Shock. Very Specific Location; Precise and Clear Pain Description; Lack of Vegetative Symptoms.
+Vegetative Symptoms: Malaise/Oppression/Syncope, Fatigue, Irritability, Pupil Dilation, Salivation Inhibition, Tachycardia, Nausea/Vomiting, Pallor, Diaphoresis, Anxiety.	+Caudal Radiation to The Corresponding Dermatome(S)
	+Pelvic Motor Dysfunction: Pelvic Organ Dysfunctions, Sexual Dysfunction, Locomotion Dysfunction

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
