# Peer review of "Neuropelveology: An Emerging Discipline for the Management of Pelvic Neuropathies and Bladder Dysfunctions through to Spinal Cord Injury, Anti-Ageing and the Mars Mission"

_jcm, 2020, doi:10.3390/jcm9103285_

Round 1

Reviewer 1 Report

This review is a discussion of neuropelveology, a medical specialty that focuses on pelvic nerve pathology. The review covers a very important topic that is well-demonstrated by the author: the treatment of chronic neuropathic pain and dysfunction. However, the manuscript has significant room for improvement, specifically in the areas of current research and discussion of the future direction of the field in the way of up-and-coming translational research instead of less applicable (but admittedly more flashy) uses in a future Mars Mission. With discussion only of GNS and LIONS procedures, this reviewer was left wondering why the field of neuropelveology neglects to acknowledge and discuss the other various forms of potentially beneficial application of neuromodulation for persons with pelvic dysfunctions. This manuscript could also benefit from editing by a native English speaker for grammar and flow.

The introduction first lays out the need for understanding of the operational functional anatomy of the pelvis, such that laparoscopic surgeons could better avoid patient consequences of postoperative pain and dysfunction. The author briefly touches on the diagnostic procedures and introduces “various medical treatments”, specifically LION procedures and GNS.

Line 30-32: The statement “Patients who presented to neuro-urologists and neurologists did not

find much help, only neuroleptic treatments but without any effort to research or treat the cause of

the symptoms.” needs to cite a reference.

Line 54-57 and 112-114: The statements “It is rare that one single person has established an entire medical discipline with such profound benefits for patients otherwise abandoned by the medical profession but this is precisely the case in neuropelveology, which was created and developed thanks to the perspicacity and perseverance of the author of this article.” and “This is why we developed the LION procedure of the sacral plexus 11,12 and then the pudendal LION procedure 13, however both techniques of implantation remain too difficult for the generalist gynaecologist trained in surgery but not in neuropelveological procedures.” are opinions that do not add academic interest and should be omitted.

The “Neuropelveology For the Management Of Chronic Neuropathic Pelvic Pain” section well-describes the diagnosis and work up of neuropathic pelvic pain from the neuropelveology perspective.

The “GNS Therapy” sections gives great detail to the surgical procedure of GNS, which allows for stimulation of the dorsal nerve of the penis/clitoris.

Line 109-112: “There is definitively a need for a more suitable alternative for selective stimulation within the pelvic cavity, a method that cannot only be reserved for experts in this field but for all gynaecologists dealing in daily practice with patients suffering from functional disorders of the bladder.” Why specifically must stimulation be within the pelvis? Recent studies (see various Herrity et al. and Gad et al. publications) have demonstrated that both epidural and transcutaneous stimulation have beneficial effects on bladder, bowel, and sexual function in persons with LUTS, as well as those with SCI.

Line 170-171: I am unsure if the author purposefully referred to the generator as “him”, but it is inappropriate to assign gender to inanimate objects in scientific writing.

There is no reference for the preliminary study of seven patients discussed in the GNS section. Without a full presentation of the methods and results, these data should not be presented here alone. Further, the authors neglect to discuss a number of other studies (that have been published and can be cited) involving GNS. See Bourbeau et al, McGee et al, Yeh et al, Brose et al, and while briefly referenced, a slightly more in depth discussion of references 14-17 would benefit this section and provide the readers with more appropriate evidence of the benefits of GNS.

Figure 6 – The figure shows a schematic of the female external genitalia; therefore, the use of the abbreviation DNP is inappropriate, as defaulting to the masculine continues the decades long trend of disregarding sex-specific differences in therapeutic treatments (which very well may be an important consideration in neuromodulation in regards to stimulation parameters). If the schematic shown is female, the pathway should be labeled as the DNC.

Lines 187-188 is a repeat of Lines 170-171.

The “LION Procedures” section is less detailed with the general procedure; however, the authors provide a link to a video, which may be helpful for clinicians.

While the clinical benefits of the LION procedure in persons with SCI is discussed, there is a lack of discussion of the potential mechanisms through which neuromodulation of the sciatic and pudendal nerves may lead to recovery of supra-spinal control. A brief statement here would be interesting.

Though exciting, the shift of discussion to voluntary locomotor function in persons with SCI shifts away from how the definition and focus of neuropelveology has previously been described. Did these patients see improvement in pelvic dysfunctions or pain? Are there other clinical studies showing improvements in pelvic health with similar stimulation? (yes). I do not think omission of the locomotor function improvements is necessary; however, the section would greatly benefit from more on-topic discussion.

Author Response

We thanks the reviewer for all his imputs and made the changes according to his recommandations.

  • The introduction first lays out the need for understanding of the operational functional anatomy of the pelvis, such that laparoscopic surgeons could better avoid patient consequences of postoperative pain and dysfunction. The author briefly touches on the diagnostic procedures and introduces “various medical treatments”, specifically LION procedures and GNS.

Answer: we completely agree and included:

Incidences of pelvic nerves pathologies are widely underestimated obviously because of lack of awareness that such lesions may exist, lack of diagnosis and acceptance, declaration and report of such lesions. The most probable reasons for omission of the pelvic nerves in medicine are the complexity of the pelvic nerve system, the difficulties of etiologic diagnosis and - probably the main reason - the limitations of access to the pelvic nerves for neurophysiologic explorations and neurosurgical treatments. Neurosurgical procedures techniques are well established in nerve lesions of the upper limb but pelvic retroperitoneal areas and surgeries to the pelvic nerves are still unusual for neurosurgeons. Few open-surgical approaches to the sacral plexus have been described by neurosurgeons for treatment for traumatic pelvic plexopathies, but these approaches are laborious and invasive, offer only limited access to the different pelvic areas and expose patients to risk of severe vascular complications. Techniques of nerves neuromodulation to control pelvic pain syndromes and dysfunctions are for the same reasons, limited to spinal cord and sacral nerves roots stimulation that restrict considerably their indications and effectiveness.  

The use of the endoscope in combination with neurofunctional surgical procedures to the pelvic nerves proved to be a decisive advantage in this development 1-4, and in fact it was the beginning of a new medical specialty, the neuropelveology 5-7. This specialty combines the knowledge required for a proper neurologic diagnosis, which is essential for an adapted treatment for intractable pelvic neuropathies. The concept of “neuropelveology”, the first medical practice focused on the pathologies of the pelvic nervous system was introduced more than twenty years ago by Possover; Since then, neuropelveology has established itself as a specialty in its own right, promulgated by the creation of the International Society of Neuropelveology in 2014.

Neuropelveology presents three consecutive aspects, the diagnostic stage followed by the therapeutic stage and the post-therapeutic follow-up of the patient, and covers for major areas:

  1. Diagnosis and treatment of Pelvic Neuropathic Pain with in particular new techniques of laparoscopic pelvic nerves decompression and neurolysis.
  2. Treatment of Pelvic Organs Dysfunctions with in particular the stimulation of the genital nerves – GNS therapy
  3. The technique of Laparoscopic Implantation Of Neuroprothesis to the pelvic nerves –LION procedure – for recovery loss functions in spinal cord injured peoples
  4. The stimulation of the pelvic autonomic nervous system for prevention and/or treatment general medical conditions such as osteoporosis, some cardio-vascular disease or control of sarcopenia – process of ageing.
  • Line 54-57 and 112-114: The statements “It is rare that one single person has established an entire medical discipline with such profound benefits for patients otherwise abandoned by the medical profession but this is precisely the case in neuropelveology, which was created and developed thanks to the perspicacity and perseverance of the author of this article.” and “This is why we developed the LION procedure of the sacral plexus 11,12 and then the pudendal LION procedure 13, however both techniques of implantation remain too difficult for the generalist gynaecologist trained in surgery but not in neuropelveological procedures.” are opinions that do not add academic interest and should be omitted.

Answer: we  agree and removed both sentneces

  • The “Neuropelveology For the Management Of Chronic Neuropathic Pelvic Pain” section well-describes the diagnosis and work up of neuropathic pelvic pain from the neuropelveology perspective.

 Answer: for more information we further included:

Intervention in area of the pelvic somatic nerves, which is covered by large vessels and a dense network of lymph nodes has hitherto been hindered by the lack of minimally invasive surgical methods. However, developments in video endoscopy enable exploration of the retroperitoneal pelvic space with access to the lumbosacral plexus and possibilities for nerve decompression and neurolysis. The most frequent aetiologies treated in neuropelveology are:

  • Sacral radiculopathy by “vascular or fibrotic entrapment” 10-12
  • Compression of the sacral plexus by hypertrophy or atypical insertion of the piriform muscle
  • Deeply infiltrating of the sacral plexus and the sciatic nerve 13-14
  • Tumor of the sacral plexus (Fig.1) 15-16 and post-surgeries pelvic neuropathies 17-18.

  • The “GNS Therapy” sections gives great detail to the surgical procedure of GNS, which allows for stimulation of the dorsal nerve of the penis/clitoris. Line 109-112: “There is definitively a need for a more suitable alternative for selective stimulation within the pelvic cavity, a method that cannot only be reserved for experts in this field but for all gynaecologists dealing in daily practice with patients suffering from functional disorders ft he bladder.” Why specifically must stimulation be within the pelvis? Recent studies (see various Herrity et al. And Gad et al. Publications) have demonstrated that both epidural and transcutaneous stimulation have beneficial effects on bladder, bowel, and sexual function in persons with LUTS, as well as those with SCI.

 Answer: we agree and made the changes: „There is definitively a need for a more suitable alternative for neuromodulative treatments,..”

  • Line 170-171: I am unsure if the author purposefully referred to the generator as “him”, but it is inappropriate to assign gender to inanimate objects in scientific writing.

 We removed the word „him“

  • There is no reference for the preliminary study of seven patients discussed in the GNS section. Without a full presentation of the methods and results, these data should not be presented here alone. Further, the authors neglect to discuss a number of other studies (that have been published and can be cited) involving GNS. See Bourbeau et al, McGee et al, Yeh et al, Brose et al, and while briefly referenced, a slightly more in depth discussion of references 14-17 would benefit this section and provide the readers with more appropriate evidence of the benefits of GNS.

 We did not give any reference because other „GNS“ technques do not focuse on stimulation oft he DNP by implanted electrodes at the genital area.

Figure 6 – The figure shows a schematic of the female external genitalia; therefore, the use of the abbreviation DNP is inappropriate, as defaulting to the masculine continues the decades long trend of disregarding sex-specific differences in therapeutic treatments (which very well may be an important consideration in neuromodulation in regards to stimulation parameters). If the schematic shown is female, the pathway should be labeled as the DNC.

In line 152 we emntione „Dorsal Nerve oft he Penis/Clitoris – DNP. This nerve is classicaly named as DNP; the terminaology DNC is unusual.

  • Lines 187-188 is a repeat of Lines 170-171.

 We agree and remove this sentence

  • The “LION Procedures” section is less detailed with the general procedure; however, the authors provide a link to a video, which may be helpful for clinicians.

 We did not described in detail the technique of LION procedure since the procedure differ from which nerve is implantated and would increase considerably the lenght oft he mansucript.

  • While the clinical benefits of the LION procedure in persons with SCI is discussed, there is a lack of discussion of the potential mechanisms through which neuromodulation of the sciatic and pudendal nerves may lead to recovery of supra-spinal control. A brief statement here would be interesting.

 We added a brief statement: The mechanism of recovery walking functions in SCI peoples after the LION procedure is still unknown. There is increasing evidence to suggest that neuromagnetic/electrical modulation promotes neuroregeneration and neural repair by affecting signaling in the nervous system, but our findings suggests that the information signals to the brain might use not only anatomical nerve pathways but also functional pathways activated by a continuous low-frequency stimulation of the low-motor neurons below the spinal cord lesion.

  • Though exciting, the shift of discussion to voluntary locomotor function in persons with SCI shifts away from how the definition and focus of neuropelveology has previously been described. Did these patients see improvement in pelvic dysfunctions or pain? Are there other clinical studies showing improvements in pelvic health with similar stimulation? (yes). I do not think omission of the locomotor function improvements is necessary; however, the section would greatly benefit from more on-topic discussion.

We added one further sentence on the LION procedure in multiple sclerosis patieints reproted by N. Lemos “The LION procedure to the pelvic somatic nerves has been further reported for treating urinary dysfunctions and improving locomotion in multiple sclerosis patients 45.”

Reviewer 2 Report

Prof. Possover introduces into the field of neuropelveology and describes impressively the LION (laparoscopic implantation of neuroprothesis) procedure and the technique of GNS (genital nerve stimulation). Both methods have been developed by him a few years ago. The LION procedure has been proven to treat chronic pelvic pain as well as functional disorders of pelvic organs such as bladder and bowel and also neuromuscular disorders such as palsy and spacicity. The GNS can improve bladder function and is convenient to apply since it only needs the surgical knowledge of TVT application. In addition, the passive electrical nerve stimulation (ENS) may contribute to anti-aging prophylaxis and prevention of muscle loss in paralyzed patients and also astronauts. The review is nicely written and the reader understands the key messages. Prof. Possover is the inventor of the field of neuropelveology and describes his findings and further suggestions impressively. Therefore, I strongly  recommend to publish the paper in its current version.

Author Response

Comments and Suggestions for Authors

Prof. Possover introduces into the field of neuropelveology and describes impressively the LION (laparoscopic implantation of neuroprothesis) procedure and the technique of GNS (genital nerve stimulation). Both methods have been developed by him a few years ago. The LION procedure has been proven to treat chronic pelvic pain as well as functional disorders of pelvic organs such as bladder and bowel and also neuromuscular disorders such as palsy and spacicity. The GNS can improve bladder function and is convenient to apply since it only needs the surgical knowledge of TVT application. In addition, the passive electrical nerve stimulation (ENS) may contribute to anti-aging prophylaxis and prevention of muscle loss in paralyzed patients and also astronauts. The review is nicely written and the reader understands the key messages. Prof. Possover is the inventor of the field of neuropelveology and describes his findings and further suggestions impressively. Therefore, I strongly  recommend to publish the paper in its current version.

Best thanks for these very nice comments